# Common Genetic Aberrations Associated with Metabolic Interferences in Human Type-2 Diabetes and Acute Myeloid Leukemia: A Bioinformatics Approach

**DOI:** 10.3390/ijms22179322

**Published:** 2021-08-28

**Authors:** Theodora-Christina Kyriakou, Panagiotis Papageorgis, Maria-Ioanna Christodoulou

**Affiliations:** 1Tumor Microenvironment, Metastasis and Experimental Therapeutics Laboratory, Basic and Translational Cancer Research Center, Department of Life Sciences, European University Cyprus, Nicosia 2404, Cyprus; T.Kyriakou@euc.ac.cy (T.-C.K.); P.Papageorgis@euc.ac.cy (P.P.); 2European University Cyprus Research Center, Nicosia 2404, Cyprus; 3Tumor Immunology and Biomarkers Laboratory, Basic and Translational Cancer Research Center, Department of Life Sciences, European University Cyprus, Nicosia 2404, Cyprus

**Keywords:** acute myeloid leukemia (AML), type-2 diabetes mellitus (T2D), metabolic pathways, single-nucleotide polymorphisms (SNPs)

## Abstract

Type-2 diabetes mellitus (T2D) is a chronic metabolic disorder, associated with an increased risk of developing solid tumors and hematological malignancies, including acute myeloid leukemia (AML). However, the genetic background underlying this predisposition remains elusive. We herein aimed at the exploration of the genetic variants, related transcriptomic changes and disturbances in metabolic pathways shared by T2D and AML, utilizing bioinformatics tools and repositories, as well as publicly available clinical datasets. Our approach revealed that rs11709077 and rs1801282, on *PPARG*, rs11108094 on *USP44*, rs6685701 on *RPS6KA1* and rs7929543 on *AC118942.1* comprise common SNPs susceptible to the two diseases and, together with 64 other co-inherited proxy SNPs, may affect the expression patterns of metabolic genes, such as *USP44*, *METAP2*, *PPARG*, *TIMP4* and *RPS6KA1*, in adipose tissue, skeletal muscle, liver, pancreas and whole blood. Most importantly, a set of 86 AML/T2D common susceptibility genes was found to be significantly associated with metabolic cellular processes, including purine, pyrimidine, and choline metabolism, as well as insulin, AMPK, mTOR and PI_3_K signaling. Moreover, it was revealed that the whole blood of AML patients exhibits deregulated expression of certain T2D-related genes. Our findings support the existence of common metabolic perturbations in AML and T2D that may account for the increased risk for AML in T2D patients. Future studies may focus on the elucidation of these pathogenetic mechanisms in AML/T2D patients, as well as on the assessment of certain susceptibility variants and genes as potential biomarkers for AML development in the setting of T2D. Detection of shared therapeutic molecular targets may enforce the need for repurposing metabolic drugs in the therapeutic management of AML.

## 1. Introduction

Type-2 diabetes mellitus (T2D) is a chronic metabolic disorder, nowadays considered a global epidemic, with ever-increasing prevalence and high cardiovascular mortality rates [1]. Metabolic disturbances in T2D are associated with chronic hyperglycemia due to deficient insulin secretion by pancreatic β-cells and decreased insulin sensitivity in the skeletal muscle, liver, and adipose tissue [2]. During the last two decades, 85 genome-wide association studies (GWAS) have revealed 1894 single-nucleotide polymorphisms (SNPs) in 1294 genes involved in the aforementioned processes [3]. Interestingly, it was recently shown that certain T2D susceptibility genes exhibit deregulated mRNA expression in the peripheral blood of patients and predisposed individuals, possibly mirroring the aberrant regulation in disease-target organs [4].

T2D also has been associated with the development of various types of human neoplasia, including both solid tumors and hematological malignancies [5]. A recent study on 804,100 new cancer patients bearing different tumor types reported that 5.7% of their development was attributable to diabetes and high body mass index (BMI) [6]. Moreover, observational and Mendelian randomization studies support a strong epidemiological link between T2D and cancer [7]. Common pathophysiological background includes: (a) risk factors such as aging, obesity and physical inactivity; (b) biological processes including hyperinsulinemia, hyperglycemia, oxidative stress and chronic low-grade inflammation and (c) molecular pathways such as the insulin/insulin-like growth factor (IGF) and interleukin (IL)-6/signal transducer and activator of transcription 3 (STAT3) axes [5]. Importantly, the first-line anti-diabetic drug metformin is known to lower the risk of cancer development in T2D patients and improve the response to anti-cancer therapies in diabetic or non-diabetic individuals bearing certain tumor types [8]. At the cellular level, the drug exerts its anti-cancer function by interfering with mitochondrial respiration and activating the AMP-activated protein kinase (AMPK) pathway [8]. At the systemic level, metformin suppresses insulin/IGF-1 and nuclear factor-κB (NF-κB) signaling pathways, downregulates the release of proinflammatory cytokines and augments CD8^+^ T cell anti-tumor responses [8].

Among hematological malignancies, acute and chronic leukemias have been associated with a previous history of T2D. A recent meta-analysis of 18 studies involving 10516 leukemia cases within a total of more than 4 million individuals with diabetes showed that the risk for the disease is increased in patients with T2D but not in patients with type 1 diabetes [9]. Especially for acute myeloid leukemia (AML), a life-threatening hematological malignancy with critical survival rates [10], it has been described that the standard incidence ratio in a cohort of 641 T2D individuals is 1.36 (95% CI: 1.26–1.47), significantly higher than in the general population [11]. Furthermore, various studies have detected BMI as an independent adverse prognostic factor for AML [12,13,14], which aggravates the relative risk for the disease in T2D [9,15]. Additionally, metformin has been associated with improved outcomes also in patients with leukemias [16]. On the other hand, in vitro studies have described that AML cells exhibit a hyper-metabolic phenotype that involves upregulations in basal and maximal respiration [17] and perturbations in glycolysis and oxidative phosphorylation processes [18,19]. These clinical and in vitro data suggest that repurposing metformin could possibly modify leukemic cells’ metabolism, indicating a promising option for the management of AML [16].

Despite the identified epidemiological association of AML with T2D, the genetic and molecular links between the two disorders remain unclear. The possible existence of common metabolic interferences that may underlie the development and perpetuation of the disease has not yet been investigated. Neither is it known whether these are attributed to aberrations in the genomic, transcriptional, or post-transcriptional level. To this end, we herein investigated a network of common genetic alterations (single-nucleotide polymorphisms, SNPs) and co-inherited variants, related mRNA deviations and pathway deregulations in the two conditions, utilizing appropriate bioinformatic tools and publicly available clinical datasets. Priority was given to the identification of gene sets and pathways associated with possible metabolic disturbances, perchance known to be related to T2D, that may control the development of AML. To the best of our knowledge, our results provide the first information regarding common genetic predisposition and connected mechanisms that may lead to the development of AML in the setting of T2D.

## 2. Results

### 2.1. Common Susceptibility SNPs in AML and T2D

Data on all SNPs associated with AML or T2D development were downloaded from the NHGRI-EBI Catalog (Appendix A). The numbers of SNPs listed and further processed were 5321 for AML and 1894 for T2D, as depicted in Figure 1A. Of these, five SNPs (rs11108094, rs1801282, rs7929543, rs11709077, rs6685701) were found to be linked with the development of both AML and T2D. All of them exerted a *p*-value for the association with either disease of <5 × 10^−8^, which was set as a threshold of significance. These five SNPs were included in the subsequent analyses of this study as significantly associated with both AML and T2D. Corresponding information on these SNPs is summarized in Table 1. In addition, information regarding their frequency in the general population is reported in Appendix A.

Two of these SNPs (rs11709077, rs1801282) lie in the *PPARG* (peroxisome proliferator-activated receptor gamma) gene, exerting the following *p*-values: for rs11709077 5 × 10^−11^ for AML and 2 × 10^−36^ for T2D, and for rs1801282 5 × 10^−11^ for AML and 2 × 10^−19^ for T2D. Another common SNP, the rs6685701, is found in the gene encoding for the ribosomal protein S6 kinase A1 (*RPS6KA1*) and exhibits a significant association with AML (*p* = 6 × 10^−18^) and T2D (*p* = 1 × 10^−08^). *USP44* (Ubiquitin Specific Peptidase 44) also bears an SNP (rs11108094) significantly related to both AML and T2D development (*p* = 2 × 10^−10^ and 6 × 10^−10^, respectively). Last, rs7929543, located in *AC118942.1* (NADPH oxidase 4 pseudogene), is also significantly associated with both AML (*p* = 7 × 10^−09^) and T2D (*p* = 2 × 10^−09^). It is important to note that all SNPs are in non-coding regions except SNP rs1801282 which is a missense variant in *PPARG*, also known as Pro12Ala. The more common C allele encodes for the Pro amino acid at the SNP position [20].

To investigate whether these genetic variants affect the expression levels of associated or other genes in disease-related tissues (adipose, skeletal muscle, liver, pancreas, whole blood), we searched for eQTLs through the GTex and Blood eQTL Browser databases [21,22]. All results obtained are reported in Table 2. Moreover, graphical data from the GTex portal are shown in Figure 1B; corresponding data from Blood eQTL Browser were not available. Rs11709077 (allele: G/A; minor allele: A) and rs1801282 (G/C; minor: G), on the *PPARG* gene, were found to affect the mRNA expression levels of *SYN2* (synapsin II) in the skeletal muscle (Figure 1B and Table 2) and whole blood (Table 2). In the skeletal muscle, the presence of the minor alleles correlates with increased *SYN2* expression (normalized effect size (NES): 0.35 and 0.36 for rs11709077 and rs1801282, respectively) (Figure 1B and Table 2), whereas in the whole blood, they are correlated with decreased levels (z-score: −3.61, for both) (Table 2). In addition, rs1801282 was found to negatively impact the expression of the *GATA3* transcription factor in whole blood (z-score = −4.54) (Table 2) and of *TIMP4* (TIMP metallopeptidase inhibitor 4) (NES = −0.21) in visceral adipose tissue (Figure 1B and Table 2). The rs11108094 variant (C/A; minor allele: A) on *USP44* was associated with decreased expression of *METAP2* (methionine aminopeptidase 2) in subcutaneous and visceral adipose tissue (NES: −0.64 and −0.55, respectively) (Figure 1B and Table 2). Finally, in visceral adipose tissue, rs6685701 (A/G; minor allele: G) in *RPS6KA1* negatively affects its own expression levels (NES: −0.099), while rs7929543 (A/C; minor allele: C) on *AC118942.1* is positively associated with the expression levels of *RP11-347H15.5* (clone-based (Vega) gene) (NES: 0.53) (Figure 1B and Table 2).

### 2.2. Proxy SNPs of the Five Common AML/T2D Susceptibility SNPs

Apart from the SNPs directly identified to be associated with a disease, other co-inherited SNPs may also lead to its development [23]. Based on this, we searched for the proxy SNPs of the five common AML/T2D susceptibility SNPs, utilizing the LDLink tool [24]. The selection criterion for a proxy SNP was to possess a squared correlation measure (R^2^) of LD greater than 0.8. Data are shown in Figure 2 and Table 3. Sixty-six (66) unique proxy SNPs that lie in the *USP44*, *METAP2*, *PPARG*, *TIMP4*, *FOLH1* (folate hydrolase 1), *AC118942*.1 and *RPS6KA1* genes were identified; some of them were detected as proxies for more than one of the five common SNPs. Through this analysis, it was also revealed that two of the common AML/T2D susceptibility genes (rs1801282, rs11709077) on the *PPARG* gene were mutual proxy SNPs (Table 3; bold/italics highlighted). Moreover, Venn diagram analysis revealed that one of the 64 SNPs (rs11519597) is an AML-specific disease susceptibility SNP, while two of them (rs71304101, rs17036160) are T2D-specific disease susceptibility SNPs (data not shown).

Furthermore, to pinpoint possible deregulation at the mRNA levels, attributed to the 64 proxy SNPs, we performed analysis using the GTex and Blood eQTL databases for the identification of eQTLs in disease-affected tissues (Table 2).

### 2.3. Common Susceptibility Genes in AML and T2D

Beyond the identification of specific genetic variants associated with both AML and T2D, we proceeded to the detection of common susceptibility genes between the two disorders. Analysis using combined data from the GWAS Catalog and the GTex portal showed that 86 genes bear SNPs that have been significantly associated with the development of both diseases, as per GWAS performed (Figure 3A). These include the five genes with common SNPs and another 81 disease-specific genes. Notably, most of the genes contain a significantly higher number of SNPs associated with AML compared to T2D (Table 4).

To investigate whether these genes comprise eGenes, which have at least one eQTL located near the gene of origin (*cis*-eQTL) acting upon them, affected by AML or T2D-specific SNPs in-disease target tissues, we searched through the GTex and eQTL Browsers. Analysis using Venn diagrams identified AML- or T2D-specific SNPs/eQTLs in certain susceptibility genes in adipose, muscle tissue, liver, pancreas and/or whole blood (Figure 3B). In adipose tissue, 6517 eQTLs on common AML/T2D susceptibility genes were detected, of which 79 were AML- and 8 T2D-specific. In skeletal muscle, 4220 were identified—28 AML- and 5 T2D-specific. In liver, 602 were detected—seven AML- and none T2D-specific. In pancreas, 3507 were found—36 AML- and 5 T2D-specific. Finally, in whole blood, 7187 were identified—55 AML- and 10 T2D-specific. A complementary analysis of the same data revealed the distribution of the AML- or T2D- SNPs/eQTLs in disease-target tissues and identified common and tissue-specific ones (Figure 3C and Table 5). All identified eQTLs affecting the 86 common disease susceptibility genes are included in Appendix A.

### 2.4. Pathway Analysis of the Proteins Encoded by the Common AML/T2D Susceptibility Genes

To investigate the possible involvement of the 86 common susceptibility genes in molecular networks correlated with both disorders, the developed gene/protein panel was further processed through the STRING and KEGG databases [25,26]. The following eGenes found to be affected by the five common susceptibility SNPs as well as by their proxies in disease-affected tissues were included in the analysis: *DHDDS* (Dehydrodolichyl Diphosphate Synthase Subunit), *GATA3*, *METAP2*, *RP11-347H15.5*, *RPS6KA1*, *SYN2*, *TIMP4*. The corresponding protein–protein interaction (PPI) network is depicted in Figure 4A. Analysis revealed that numerous proteins of the above set are significantly involved in metabolic pathways, including pyrimidine, purine, choline metabolism, mTOR, AMPK, PI_3_K-Akt and insulin signaling, as well as pathways deposited as related to AML (FDR < 0.05 for all) (Figure 4B and Table 6).

Differently colored nodes designate various genes/proteins involved in one or more pathways. Edges represent protein–protein associations—either known interactions, predicted interactions or other associations. All regulated pathways revealed in this analysis are included in Appendix A.

### 2.5. Investigation of Aberrant mRNA Expression of T2D-Deregulated Genes in an AML Cohort

The second aim of the study was to investigate the possible deregulation of T2D-related metabolic mechanisms in AML patients. To this end, we selected a panel of genes previously reported to be deregulated in T2D patients [4] (*CAPN10*, *CDK5*, *CDKN2A*, *IGF2BP2*, *KCNQ1*, *THADA*, *TSPAN8*) and explored their mRNA levels in peripheral blood samples from AML- versus non-cancerous individuals utilizing RNAseq data and the TNMplot web tool [27]. Significantly increased mRNA levels of *CAPN10*, *CDK5*, *CDKN2A*, *IGF2BP2* and THADA, as well as significantly decreased levels of *KCNQ1* and *TSPAN8*, were found in 151 AML patients compared to 407 normal individuals tested (Mann–Whitney *p* < 0.0004 for all). The percentage (%) of AML samples that displayed up- or downregulated expression for each of the above genes, at each of the four quantile cut-off values (minimum, 1st quartile, median, 3rd quartile, maximum), as well as the specificity (the ratio of the number of AML samples to the sum of AML and non-cancerous samples over or below each given cut-off), are depicted in Figure 5.

To search for AML-specific SNPs on these deregulated genes, we used data obtained from the NHGRI-EBI Catalog of GWAS. It was found that rs10832134 (chromosomal location: 11:2481256), rs12576156 (11:2477588) and rs11523905 (11:2477029) variants lie in the *KCNQ1* (*p =* 3 × 10^−15^ for all), while the rest of the deregulated genes have not been identified to bear AML-related SNPs. Investigation for their proxies revealed three proxy SNPs (rs12574553, rs757092, rs7126330) for rs10832134 and five proxy SNPs (rs73419519, rs7937273, rs7928116, rs179395, rs7542142) for rs12576156, all of them in *KCNQ1*. No proxies were found for rs11523905 (data not shown). Out of these, the proxy SNP rs12574553 (allele C/T) consists of an eQTL for *KCNQ1*; the minor allele leads to the downregulation of mRNA levels in whole blood [21].

## 3. Discussion

Today, there is a well-accepted epidemiological link between T2D and cancer development [5]. However, in other types of human neoplasia, the association between T2D and hematological malignancies is less explored. Among them, AML represents one of the most intriguing morbidities for further investigation due to its increasing rates and relatively poor prognosis and response to treatment [10,28]. Accumulating clinical evidence connecting metabolic syndrome parameters (including BMI and T2D) to AML [9,11,12,13,14,15,16], together with corresponding in vitro data [17,18,19], highlights the need for investigation of the underlying mechanisms implicating genetic predisposition, which may regulate metabolic abnormalities.

In this study, we first aimed at the description of the possible common genetic background shared by the two disorders. Processing of the thousands of AML- and T2D-associated SNPs deposited in the GWAS NHGRI-EBI Catalog uncovered five SNPs that are significantly linked to both diseases (Table 1). Two of them (rs11709077, rs1801282) lie in the *PPARG* gene, the first gene reproducibly associated with T2D [29,30]. The gene encodes for the PPAR-γ receptor, a molecular target of thiazolidinediones (insulin-sensitizing antidiabetic drugs); gene variants affecting its transcription levels in adipose tissue are associated with insulin sensitivity [29,30]. Although there are no data directly linking *PPARG* with AML, it is worth mentioning that the protein is implicated in the TGF-beta and mTOR signaling pathways, both associated with cancer development [31,32,33]. Our analyses also indicated that rs11709077 and rs1801282 on *PPARG* negatively affect the expression of *SYN2* (Synapsin II) in skeletal muscle and in whole blood (Table 2, Figure 1); however, there is not yet any evidence connecting *SYN2* with T2D or AML.

Another common SNP, which is a missense variant rs1801282, was found to negatively regulate the expression of the tissue inhibitor of metalloproteinases 4 (*TIMP4*) in visceral adipose tissue. The TIMP family has been associated with several cancers [34], but no information about its relation to T2D is available yet. Another interesting observation regards the negative impact of rs1801282 on *GATA3* in whole blood. GATA3 is a transcription factor with a multi-faceted role in hematopoiesis [35], while related genetic and epigenetic aberrations are strongly associated with AML development, prognosis and response to therapy [36,37]. Regarding T2D, GATA3 is considered an anti-adipogenic factor and a potential molecular therapeutic target for insulin resistance, through restoration of adipogenesis and amelioration of inflammation [38,39].

Rs6685701, located in the gene encoding for the ribosomal protein S6 kinase A1 (*RPS6KA1* or *P90S6K*), was found to be associated with its lower expression levels in visceral adipose tissue. The protein belongs to the family of serine/threonine kinases that govern various cellular processes, and it acts downstream of ERK (MAPK1/ERK2 and MAPK3/ERK1) signaling [33]. In murine models of T2D, *RPS6KA1* has been implicated in impaired glucose homeostasis in β-pancreatic, muscle and liver cells [40,41], which is improved upon sitagliptin (DPP-4 inhibitor; antidiabetic drug) administration [42]. Using an in vivo model of leukemia, RPS6KA1 has been shown to promote the self-renewal of hematopoietic stem cells and disease progression through the regulation of the mTOR pathway [43]. More importantly, it was very recently reported that RPS6KA1 may be a strong indicator of overall survival in AML patients, while aberrations in the miR-138-5p/RPS6KA1 axis are associated with poor prognosis among patients [44].

The rs11108094 in *USP44* (ubiquitin-specific peptidase 44) was also recognized as a common susceptibility variant for AML and T2D, which acts as an eQTL downregulating the expression of *METAP2* (methionyl aminopeptidase 2) in subcutaneous and adipose tissue. The USP44 protein is implicated in protein metabolism and ubiquitin-mediated proteasome-dependent proteolysis. More importantly, *METAP2* is involved in the metabolism of fat-soluble vitamins [33]. Its inhibition results in weight loss in obese rodents, dogs and humans and has been proposed as a therapeutic target against obesity [45]. On the other hand, METAP2 inhibitors have been shown to induce apoptosis in leukemic cell lines [46], which renders them potent therapeutic agents also for leukemia. Lastly, the rs7929543 variant on the *AC118942.1* pseudogene was identified as an eQTL influencing the expression of the *RP11-347H15.5* pseudogene in visceral adipose tissue. The involvement of this deregulation in possible pathogenetic processes for both diseases might be part of the complex underlying genetic–molecular mechanisms.

To describe the network of genetic variants’ inheritance more extensively, we developed a panel of 64 unique proxy SNPs associated with the five common AML/T2D ones (Table 2). Interestingly, these proxies are found to lie within and/or be eQTLs for the aforementioned genes (*PPARG*, *SYN2*, *TIMP4*, *GATA3*, *RPS6KA1*, *USP44, METAP2, AC118942.1, RP11-347H15.5*) in disease-target tissues. A new eGene added to the panel was *DHHS*, which is downregulated in whole blood by SNPs on *RP11-347H15.5*. The gene encodes for the dehydrodolichyl diphosphate synthase subunit and is involved in pathways of protein metabolism and in N-glycan biosynthesis [33]. However, no direct data connecting the gene with neoplasias or diabetes have been reported to date.

Next, we identified a panel of 86 common AML/T2D susceptibility genes using the GWAS NHGRI-EBI Catalog (Figure 3). Several SNPs specific for each disease were found to impact the expression patterns of some of these common susceptibility genes in affected tissues, suggesting their possible functional involvement in disease development (Table 5). Pathway analysis revealed that the AML/T2D gene set regulates a series of metabolic pathways, with the highest significance observed for pyrimidine and purine metabolism. Although neither AML or T2D is purely a disorder of pyrimidine and/or purine metabolism, there are data supporting their implication in the development of each disease. The insulin effect on their regulation in diabetic liver is knowledge obtained decades ago [47,48]. Nevertheless, it was very recently described that the signatures of purine metabolites, including betaine metabolites, branched-chain amino acids, aromatic amino acids, acylglycine derivatives and nucleic acid metabolites, are associated with hyperglycemia or insulin resistance [49,50]. While there is no recent evidence regarding a possible role for purine and pyrimidine metabolites in leukemia, older studies support the notion that reciprocal alterations in the phenotype of specific enzymes may occur in leukemia cells [51,52].

Choline metabolism is another pathway that emerged through gene set enrichment analysis. Indeed, its upregulation in malignant transformation is well described [53], while the serum metabolomic signature of AML patients includes parameters of aberrant choline metabolism [54]. A group of metabolic pathways, including those of carbohydrates, lipids, nucleotides, amino acids, glycans, cofactors, vitamins, biosynthesis of terpenoids, polyketides and other secondary metabolites [25], as well as signaling pathways related to metabolic disturbances and the development of neoplasia and T2D, such as mTOR, AMPK, PI_3_K-Akt and insulin signaling pathways, were also among the ontologies significantly regulated by the AML/T2D gene set. Analysis also revealed an association with a pathway category deposited as “Acute Myeloid Leukemia”, which refers to ERK, PI_3_K and JAK-STAT signaling and transcription regulation pathways including mutated RUNX1 and the fusion genes AML1-ETO, PML-RARA and PLZF-RARA [33].

Finally, exploration through clinical datasets revealed that certain T2D-related genes, previously shown to be deregulated in T2D individuals [4], also exhibit deviated transcriptomic levels in AML patients. Expression levels of *THADA* (thyroid adenoma-associated protein), *IGF2BP2* (insulin-like growth factor 2 mRNA binding protein 2), *CDKN2A* (cyclin-dependent kinase inhibitor 2A) and *CDK5* (cyclin-dependent kinase 5) were upregulated, while levels of *KCNQ1* (potassium voltage-gated channel subfamily Q member 1) were downregulated in the peripheral blood of AML patients compared to normal subjects. *IGF2BP2*, *CDKN2A*, *CDK5* and *KCNQ1* are known to be implicated in the mass development, proliferation, and insulin secretory function of β-cells, and in metabolic processes in T2D-affected tissues [3,20,55,56]. As for *THADA*, despite its susceptibility to T2D, there are no data yet related to its involvement in the disease’s pathogenesis and/or metabolic pathways [4]. However, chromosomal aberrations engaging this gene are observed in benign thyroid adenomas [57]. *CAPN10* (calpain 10) shows increased whereas *TSPAN8* (Tetraspanin 8) exhibits decreased mRNA levels in AML versus non-cancerous individuals, a trend opposite to what was observed in T2D versus healthy subjects. CAPN10 plays important roles in the translocation of glucose transporter 4 (GLUT4), secretion of insulin and apoptotic processes in pancreatic cells [57], while *TSPAN8* has been described as a prognostic indicator for patients with certain solid tumors [58,59], but not for hematological malignancies.

In summary, this study provides, for the first time, evidence for a strong genetic network that is related to aberrations in metabolic processes and molecular pathways, shared between AML and T2D. Even though the metabolic vulnerability of AML cells and aberrant metabolic pathways observed in AML patients [54,60] have increasingly gained the attention of the research community, the genetic background leading to these metabolic disturbances had not yet been investigated. Data emerging from our study revealed that: (i) specific genetic variants (SNPs) associated with both AML and T2D, as well as their co-inherited proxy SNPs, mostly specific for each disease rather than common, can alter the gene expression patterns in disease-target tissues; (ii) common susceptibility genes and genes with altered expression may be linked to the development of AML or T2D through common (such as *PPARG*) or different mechanisms (such as *GATA3*) and (iii) common susceptibility genes can regulate metabolic pathways, which may be implicated in the pathogenetic mechanisms leading to the development of the two disorders. It should be noted, however, that the study has certain limitations, including that it exclusively analyzed in silico data and the fact that other parameters affecting the gene expression, such as epigenetic mechanisms, were not explored. Moreover, in the case of certain genes and their SNPs, i.e., those of *PPARG* and *GATA3*, their specific implication in AML and/or T2D development is not well documented. Therefore, it is yet difficult to provide a plausible explanation regarding their possible impact as risk factors for AML in the context of T2D. Lastly, it needs to be clarified that, although some of the reported SNPs are associated with certain genes involved in AML (such as *RPS6KA1* and *METAP2*), the latter are not considered driver genes for AML initiation.

Despite these limitations, significant evidence emerging from this study can be further explored in future basic and clinical studies. For example, the common susceptibility genes revealed can be evaluated for their potential to serve as prognostic biomarkers of AML development in cohorts of T2D individuals. Moreover, in depth exploration of the described metabolic pathways and involved genes may lead to a better understanding of the pathogenetic basis of the increased risk for AML development observed in individuals with T2D. Finally, detailed investigation of the common therapeutic targets identified may suggest that repurposing of metabolic drugs (i.e., DPP-4 inhibitor targeting RPS6KA1 or thiazolidinediones targeting PPAR-γ) could be exploited as novel therapeutic strategies to enhance the anti-leukemic armamentarium.

## 4. Materials and Methods

### 4.1. Study Design

Our study was performed in two axes. (A) Detection of common genetic variants and deregulated pathways in T2D and AML: We first created a panel of SNPs associated with AML or T2D, upon an in-depth search in the NHGRI-EBI Catalog of published GWAS [3], to detect common disease susceptibility genes. Their proxy SNPs were also detected using the LDLink web tool [24]. For the possible impact of the common susceptibility SNPs and their proxies on gene mRNA expression, a combined search in the Genotype-Tissue Expression (GTEx) project [21] and the Blood eQTL Browser [22] was performed. Moreover, a panel of mutual genes bearing common or disease (AML or T2D)-specific genes were processed through pathway analysis using the STRING (Search Tool for the Retrieval of Interacting Genes/Proteins) database [26], to reveal associated molecular networks and biological processes. (B) Investigation of possible deregulated expression of T2D susceptibility genes in AML cohorts: A panel of T2D susceptibility genes that were previously described to exert aberrant mRNA levels in diabetic patients was explored for their possible deregulated expression also in AML patients, using the TNMplot tool [27].

### 4.2. Development of the AML and T2D Susceptibility SNP Panels and Detection of Common SNPs

The panels of total susceptibility genes specific for AML and T2D were developed upon an in-depth search in the NHGRI-EBI GWAS Catalog [3]. All populations were considered for assessment. Common disease susceptibility genes were detected, generating Venn diagrams with the Draw-Venn-Diagrams online tool (http://bioinformatics.psb.ugent.be/webtools/Venn/) (May 2021). A genome-wide statistically significant *p*-value lower than or equal to 5 × 10^−8^ was applied to detect the SNPs that were significantly associated with the diseases. Data regarding the prevalence of the SNPs of interest in the general population were obtained from the gnomAD browser [61].

### 4.3. Detection of Proxy SNPs

Proxy SNPs of disease susceptibility SNPs of interest were detected utilizing the LDLink tool [24]. LDLink interactively explores proxy and putatively functional variants/SNPs for a query/tag variant (±500 kilobases). The tool provides information about: (A) a squared correlation measure (R^2^) of linkage disequilibrium (LD); proxy SNPs are considered those having ≥80% possibility of coinheritance with the tag SNP, which equals to a R^2^ value ≥ 0.8, and (b) the combined recombination rate (cM/Mb) from HapMap; the recombination rate is the rate at which the association between the two loci is changed. It combines the genetic (cM) and physical positions (Mb) of the marker by an interactive plot.

### 4.4. Detection of Expression Quantitative Trait Loci (eQTLs)

Expression quantitative trait loci (eQTLs), which explain variations in mRNA expression levels, related to the SNPs of interest were explored utilizing the GTEx portal and the Blood eQTL Browser [21,22]. Analysis was focused on the expression patterns in the total target tissues of the two diseases (as per their availability in the databases). These included adipose tissue (subcutaneous, visceral), skeletal muscle, liver, pancreas and whole blood.

### 4.5. Pathway Analysis

Analysis through the STRING [26] and Kyoto Encyclopedia of Genes and Genomes (KEGG) [25] databases was performed to detect protein–protein interactions possibly regulated by a panel including: (i) proteins encoded by genes that bear disease susceptibility SNPs in both AML and T2D as well as (ii) proteins encoded by genes that are commonly affected by different AML-specific and T2D-specific SNPs. To filter significantly regulated pathways, a false discovery rate (FDR) <0.05 was set as cut-off.

### 4.6. Investigation of the Expression Patterns of T2D-Deregulated Genes in AML Clinical Cohorts

To explore possible variations in the mRNA expression levels of previously described T2D-deregulated genes [4] in patients with AML, the TNMplot tool was used [27]. In more detail, analysis processed whole-exome sequencing data from 151 AML patients versus 407 non-cancerous individuals, available in the database. The tool compared the expression levels of each gene in the two groups using the Mann–Whitney non-parametric test, reporting the *p*-value of significance and the fold-change between groups. Other information included (a) the percentage (%) of AML samples that exerted up- or downregulated expression of query genes compared to non-cancerous samples, at each of the four quantile cut-off values (minimum, 1st quartile, median, 3rd quartile, maximum), and (b) the specificity, defined as the ratio of the number of AML samples to the sum of AML and non-cancerous samples over or below each given cut-off.

## Figures and Tables

**Figure 1 ijms-22-09322-f001:**
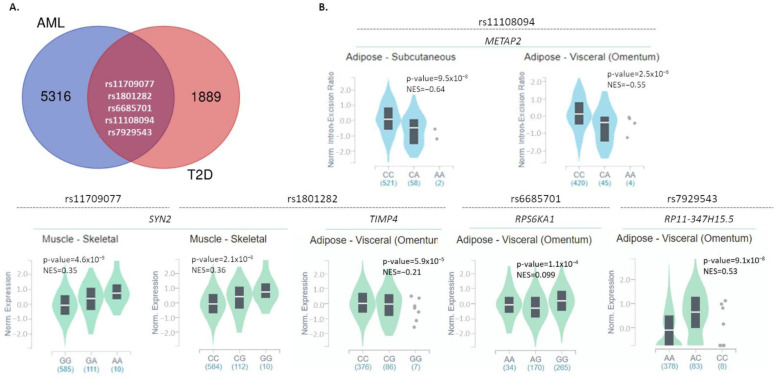
Common SNPs between AML and T2D and their impact on gene expression in disease-associated tissues. (**A**) Venn diagrams reporting the number of common and specific SNPs significantly associated with AML or T2D, based on data downloaded from the NHGRI-EBI GWAS Catalog. (**B**) Violin plots depicting the impact of the five common SNPs on the expression levels of associated or other genes, in disease-associated tissues (subcutaneous or visceral adipose tissue, skeletal muscle, liver, pancreas, whole blood) (GTex portal, May 2021). NES: normalized effect size.

**Figure 2 ijms-22-09322-f002:**
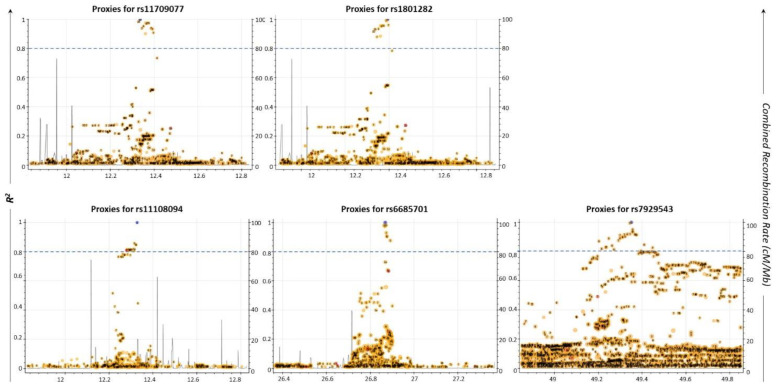
Regional LD plots of five commonly associated SNPs generated using the LDLink web tool (May 2021). Each dot represents the pairwise LD level between two individual SNPs. X-axis depicts the chromosomal coordinates. Left y-axis represents the pairwise R^2^ value with the query variant; R^2^ threshold greater than or equal to 0.8 was considered as a cut-off for selected proxies (blue dashed line). Right y-axis indicates the combined recombination rate (cM/Mb) from HapMap. Recombination rate is the rate at which the association between the two loci is changed. It combines the genetic (cM) and physical positions (Mb) of the marker by an interactive plot.

**Figure 3 ijms-22-09322-f003:**
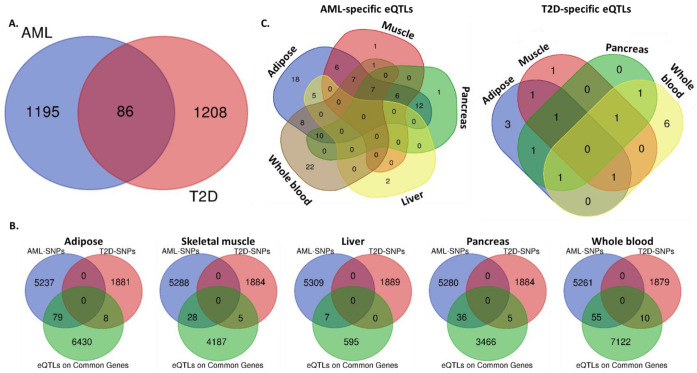
Common and disease-specific SNPs and eQTLs per target tissue. Venn diagrams reporting: (**A**) the number of common and disease-specific susceptibility genes between AML and T2D, (**B**) the numbers of AML- or T2D-specific SNPs that act as eQTLs upon the expression of common AML/T2D susceptibility genes, in adipose, skeletal muscle, liver, pancreas and whole blood, (**C**) the number of tissue-specific and common AML- or T2D- SNPs. Analysis was performed combining data from the NHGRI-EBI Catalog of GWAS and GTex portal.

**Figure 4 ijms-22-09322-f004:**
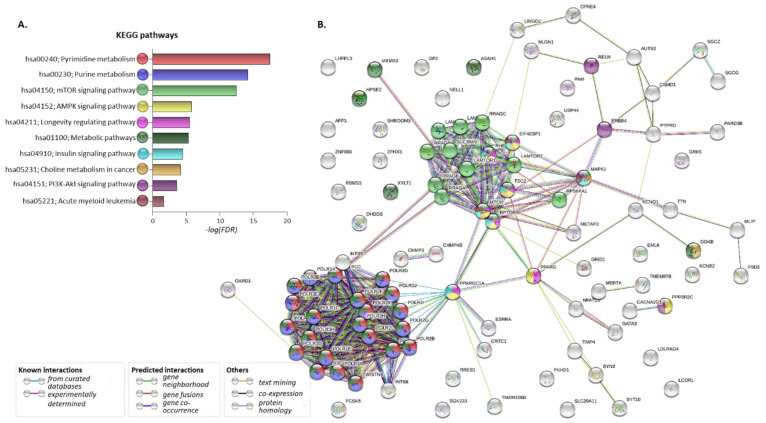
Pathways and protein–protein interactions regulated by the common AML/T2D-related genes. (**A**). Pathways enriched upon gene set analysis of 86 AML/T2D common susceptibility genes plus the seven eGenes affected by the five common AML/T2D susceptibility genes and their proxies, using KEGG database. (**B**). Protein–protein interaction (PPI) network developed upon processing the set in the STRING database. Different genes/proteins involved in different (one or more) pathways are designated by the differently colored nodes. Edges represent protein–protein associations—either known interactions, predicted interactions or other associations.

**Figure 5 ijms-22-09322-f005:**
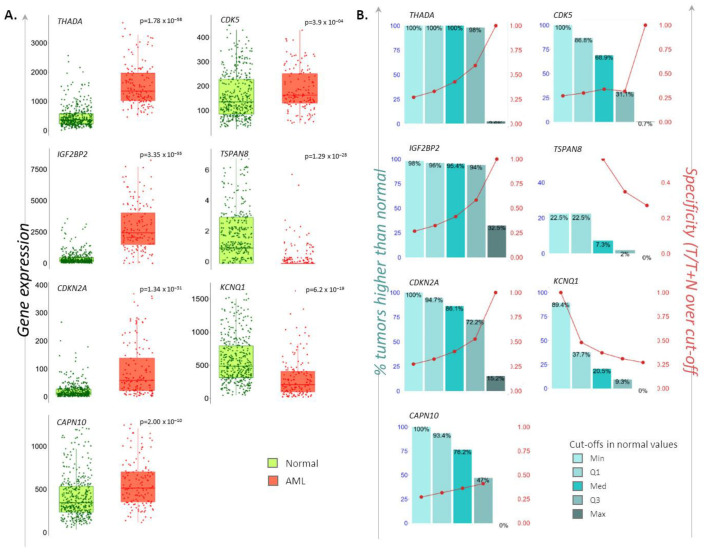
Differential expression levels of T2D-related genes in AML individuals. (**A**). Dot-plot/whisker bars depicting the differential mRNA levels of the *CAPN10*, *CDK5*, *CDKN2A*, *IGF2BP2*, *KCNQ1*, *THADA*, *TSPAN8* T2D susceptibility genes in AML patients. *P*-values of significance as obtained by Mann–Whitney test are reported. (**B**). Bar diagrams showing the: (i) percentage (%) of AML samples that possesses higher or lower of each gene-of-interest compared to non-cancerous samples, at each of the four quantile cut-off values (minimum, 1st quartile, median, 3rd quartile, maximum) (left y-axis), and (ii) specificity defined as the ratio of the number of AML samples to the sum of AML and non-cancerous samples over or below each given cut-off (right y-axis).

**Table 1 ijms-22-09322-t001:** Information about the five common SNPs associated with both AML and T2D, as obtained upon search in the NHGRI-EBI Catalog of genome-wide association studies (GWAS) (May 2021) [3]. Variant ID, chromosomal location, cytogenetic region, mapped genes, risk alleles, *p*-values detected in each study, study accession numbers and the corresponding traits are reported.

SNP	Chromosomal Location	Cytogenetic Region	Mapped Gene	Risk Allele	*p*-Value	Study Accession Number	Trait
rs11709077	3:12295008	3p25.2	*PPARG*	G	2 × 10^−36^	GCST009379	T2D
					1 × 10^−8^	GCST005047	
				A	5 × 10^−11^	GCST008413	AML
rs1801282	3:12351626	3p25.2	*PPARG*	C	3 × 10^−19^	GCST007516	T2D
					1 × 10^−17^	GCST007515	
					1 × 10^−12^	GCST005047	
					5 × 10^−12^	GCST007517	
				G	2 × 10^−14^	GCST004894	
					2 × 10^−19^	GCST004894	
					5 × 10^−11^	GCST008413	AML
rs6685701	1:26542148	1p36.11	*RPS6KA1*	G	6 × 10^−18^	GCST008413	T2D
					1 × 10^−8^	GCST010555	
				A	1 × 10^−10^	GCST008413	AML
rs11108094	12:95534337	12q22	*USP44*	C	1 × 10^−10^	GCST010557	T2D
					1 × 10^−10^	GCST010555	
					2 × 10^−10^	GCST008413	AML
rs7929543	11:49329474	11p11.12	*AC118942*.1	C	2 × 10^−9^	GCST006867	T2D
				A	7 × 10^−9^	GCST008413	AML
					6 × 10^−6^	GCST008413	

**Table 2 ijms-22-09322-t002:** eQTL associated with the five common disease susceptibility SNPs described in AML and/or T2D target tissues, as well as with their 64 proxies, as deposited in the GTEx project and Blood eQTL Browser. The SNP ID, SNP alleles, associated and affected genes and tissue(s), as well as corresponding *p*-values and the effect sizes, are reported.

SNP	Associated Gene	SNP Alleles	Affected Gene	Tissue	*p*-Value	Effect Size	Database
Five (5) common AML/T2D susceptibility SNPs							
rs11108094	*USP44*	C/A	*METAP2*	Subcutaneous adipose	9.50 × 10^−8^	NES = −0.64	GTEx project
				Visceral adipose	2.50 × 10^− 6^	NES = −0.55	GTEx project
rs11709077	*PPARG*	G/A	*SYN2*	Whole blood	3.09 × 10^−4^	Z-score = −3.61	Blood eQTL Browser
				Skeletal muscle	5.90 × 10^−5^	NES = −0.21	GTEx project
rs1801282	*PPARG*	G/C	*GATA3*	Whole blood	5.70 × 10^−6^	Z-score = −4.54	Blood eQTL browser
			*SYN2*	Whole blood	3.09 × 10^−4^	Z-score = −3.61	Blood eQTL browser
				Skeletal muscle	2.10 × 10^−8^	NES = 0.36	GTEx project
			*TIMP4*	Visceral adipose	5.90 × 10^−5^	NES = −0.21	GTEx project
rs6685701	*RPS6KA1*	A/G	*RPS6KA1*	Visceral adipose	1.10 × 10^−4^	NES = −0.099	GTEx project
rs7929543	*AC118942.1*	A/C	*RP11-347H15.5*	Visceral adipose	9.10 × 10^−8^	NES = 0.53	GTEx project
Sixty-four (64) proxies of the five common AML/T2D susceptibility SNPs							
rs10839264	*FOLH1, AC118942.1*	C/T	*RP11-347H15.5*	Visceral adipose	7.90 × 10^−8^	NES = 0.51	GTex project
rs10859889	*USP44, METAP2*	A/T	*METAP2*	Subcutaneous adipose	5.20 × 10^−8^	NES = −0.65	GTEx project
				Visceral adipose	2.30 × 10^−6^	NES = −0.54	
rs11040352	*FOLH1, AC118942.1*	A/C	*RP11-347H15.5*	Visceral adipose	5.10 × 10^−13^	NES = 0.69	GTex project
rs11040365	*FOLH1, AC118942.1*	C/A	*RP11-347H15.5*	Visceral adipose	1.40 × 10^−11^	NES = 0.65	GTex project
rs11108070	*USP44*	T/A	*METAP2*	Subcutaneous adipose	5.20 × 10^−8^	NES = −0.65	GTEx project
				Visceral adipose	2.30 × 10^−6^	NES = −0.54	
rs11108072	*USP44, METAP2*	T/C	*METAP2*	Subcutaneous adipose	5.20 × 10^−8^	NES = −0.65	GTEx project
				Visceral adipose	2.30 × 10^−6^	NES = −0.54	
rs11108076	*USP44, METAP2*	G/A	*METAP2*	Subcutaneous adipose	5.20 × 10^−8^	NES = −0.65	GTEx project
				Visceral adipose	2.30 × 10^−6^	NES = −0.54	
rs11108079	*USP44, METAP2*	G/A	*METAP2*	Subcutaneous adipose	5.20 × 10^−8^	NES = −0.65	GTEx project
				Visceral adipose	2.30 × 10^−8^	NES = −0.54	
rs11108086	*USP44*	T/C	*METAP2*	Subcutaneous adipose	5.20 × 10^−8^	NES = −0.65	GTEx project
				Visceral adipose	1.60 × 10^−6^	NES = −0.56	
rs11108087	*USP44*	A/G	*METAP2*	Subcutaneous adipose	9.50 × 10^−8^	NES = −0.64	GTEx project
				Visceral adipose	1.70 × 10^−6^	NES = −0.56	
rs11519597	*USP44, METAP2*	T/C	*METAP2*	Subcutaneous adipose	5.20 × 10^−8^	NES = −0.65	GTEx project
				Visceral adipose	2.30 × 10^−6^	NES = −0.54	
rs11522874	*USP44, METAP2*	G/A	*METAP2*	Subcutaneous adipose	5.20 × 10^−8^	NES = −0.65	GTEx project
				Visceral adipose	2.30 × 10^−6^	NES = −0.54	
rs11580180	*RPS6KA1*	A/G	*RPS6KA1*	Visceral adipose	1.40 × 10^−4^	NES = 0.098	GTEx project
rs11603576	*FOLH1, AC118942.1*	G/A	*RP11-347H15.5*	Visceral adipose	9.10 × 10^−8^	NES = 0.53	GTEx project
rs11607791	*FOLH1, AC118942.1*	T/C	*RP11-347H15.5*	Visceral adipose	7.90 × 10^−8^	NES = 0.51	GTEx project
rs11709077	*PPARG*	G/A	*SYN2*	Whole blood	3.09 × 10^−4^	Z-score = −3.61	Blood eQTL Browser
				Skeletal muscle	4.60 × 10^−9^	NES = 0.35	GTEx project
rs11712037	*PPARG, TIMP4*	C/G	*TIMP4*	Visceral adipose	7.30 × 10^−5^	NES = −0.21	GTEx project
				Skeletal muscle	2.20 × 10^−9^	NES = 0.35	
rs12146719	*USP44, METAP2*	C/A	*METAP2*	Subcutaneous adipose	5.20 × 10^−8^	NES = −0.65	GTEx project
				Visceral adipose	2.30 × 10^−6^	NES = −0.54	
rs12369757	*USP44*	G/A	*METAP2*	Subcutaneous adipose	5.20 × 10^−8^	NES = −0.65	GTEx project
				Visceral adipose	2.30 × 10^−6^	NES = −0.54	
rs13064760	*PPARG*	T/C	*SYN2*	Whole blood	2.55 × 10^−4^	Z-score = −3.66	Blood eQTL Browser
				Skeletal muscle	4.10 × 10^−9^	NES = 0.35	GTEx project
			*TIMP4*	Visceral adipose	7.50 × 10^−5^	NES = −0.21	GTEx project
rs13083375	*PPARG*	G/T	*SYN2*	Skeletal muscle	4.10 × 10^−9^	NES = 0.35	GTEx project
			*TIMP4*	Visceral adipose	7.50 × 10^−5^	NES = −0.21	
rs143400372	*USP44*	G/GA	*METAP2*	Subcutaneous adipose	9.50 × 10^−8^	NES = −0.64	GTEx project
				Visceral adipose	2.50 × 10^−6^	NES = −0.55	
rs150732434	*PPARG, TIMP4*	TG/T	*TIMP4*	Visceral adipose	7.50 × 10^−5^	NES = −0.21	GTEx project
			*SYN2*	Skeletal muscle	4.10 × 10^−9^	NES = 0.35	
rs17036160	*PPARG, TIMP4*	C/T	*TIMP4*	Visceral adipose	8.50 × 10^−5^	NES = −0.21	GTEx project
			*SYN2*	Skeletal muscle	6.50 × 10^−9^	NES = 0.34	
rs1801282	*PPARG*	G/C	*GATA3*	Whole blood	5.70 × 10^−6^	Z-score = −4.54	Blood eQTL Browser
			*SYN2*	Whole blood	3.09 × 10^−4^	Z-score = −3.61	Blood eQTL Browser
				Skeletal muscle	2.10 × 10^−8^	NES = 0.36	GTEx project
			*TIMP4*	Visceral adipose	5.90 × 10^−5^	NES = −0.21	
rs1843628	*FOLH1, AC118942.1*	A/G	*RP11-347H15.5*	Visceral adipose	3.40 × 10^−9^	NES = −0.55	GTEx project
rs1880436	*FOLH1, AC118942.1*	A/G	*RP11-347H15.5*	Visceral adipose	2.70 × 10^−9^	NES = 0.55	GTEx project
rs2012444	*PPARG*	C/T	*SYN2*	Skeletal muscle	4.10 × 10^−9^	NES = 0.35	GTEx project
			*TIMP4*	Visceral adipose	7.50 × 10^−5^	NES = −0.21	
rs2278978	*RPS6KA1*	G/A	*RPS6KA1*	Whole blood	1.96 × 10^−4^	Z-score = −3.72	Blood eQTL Browser
			*DHDDS*	Whole blood	2.41 × 10^−3^	Z-score = −3.03	Blood eQTL Browser
rs2305293	*USP44, METAP2*	C/T	*METAP2*	Subcutaneous adipose	5.20 × 10^−8^	NES = −0.65	GTEx project
				Visceral adipose	2.30 × 10^−6^	NES = −0.54	
rs35000407	*PPARG, TIMP4*	T/G	*TIMP4*	Visceral adipose	7.50 × 10^−5^	NES = −0.21	GTEx project
			*SYN2*	Skeletal muscle	4.60 × 10^−9^	NES = 0.35	
rs35788455	*PPARG*	CTTG/C	*SYN2*	Skeletal muscle	1.80 × 10^−9^	NES = 0.36	GTEx project
			*TIMP4*	Visceral adipose	8.20 × 10^−5^	NES = −0.21	
rs4443935	*RPS6KA1*	G/A	*RPS6KA1*	Whole blood	2.45 × 10^−4^	Z-score = −3.67	Blood eQTL Browser
rs4684847	*USP44, METAP2*	C/T	*TIMP4*	Visceral adipose	8.20 × 10^−5^	NES = −0.21	GTEx project
			*SYN2*	Skeletal muscle	1.80 × 10^−9^	NES = 0.36	
rs4762563	*USP44, METAP2*	G/C	*METAP2*	Subcutaneous adipose	5.20 × 10^−8^	NES = −0.65	GTEx project
				Visceral adipose	2.30 × 10^−6^	NES = −0.54	
rs61939476	*USP44, METAP2*	A/C	*METAP2*	Subcutaneous adipose	5.20 × 10^−8^	NES = −0.65	GTEx project
				Visceral adipose	2.30 x 10^−6^	NES = −0.54	
rs61939479	*USP44, METAP2*	C/T	*METAP2*	Subcutaneous adipose	5.20 × 10^−8^	NES = −0.65	GTEx project
				Visceral adipose	1.60 × 10^−6^	NES = −0.54	
rs61939481	*USP44*	T/C	*METAP2*	Subcutaneous adipose	9.50 × 10^−8^	NES = −0.64	GTEx project
				Visceral adipose	6.40 × 10^−6^	NES = −0.52	
rs71304101	*PPARG, TIMP4*	G/A	*TIMP4*	Visceral adipose	5.80 × 10^−5^	NES = −0.21	GTEx project
			*SYN2*	Skeletal muscle	9.30 × 10^−10^	NES = 0.36	
rs737465	*RPS6KA1*	C/T	*DHDDS*	Whole blood	1.88 x 10^−3^	Z-score = −3.11	Blood eQTL Browser
			*RPS6KA1*	Whole blood	2.04 × 10^−4^	Z-score = −3.71	Blood eQTL Browser
				Visceral adipose	1.40 × 10^−4^	NES = 0.098	GTex project
rs75781920	*FOLH1, AC118942.1*	T/G	*RP11-347H15.5*	Visceral adipose	2.70 × 10^−9^	NES = 0.55	GTex project
rs76218798	*FOLH1, AC118942.1*	T/C	*RP11-347H15.5*	Visceral adipose	7.90 × 10^−8^	NES = 0.51	GTex project
rs76427006	*FOLH1, AC118942.1*	T/A	*RP11-347H15.5*	Visceral adipose	2.70 × 10^−9^	NES = 0.55	GTex project
rs79067108	*USP44*	GCT/G	*METAP2*	Subcutaneous adipose	5.20 × 10^−8^	NES = −0.65	GTEx project
				Visceral adipose	2.30 × 10^−6^	NES = −0.54	

**Table 3 ijms-22-09322-t003:** Summary of the proxy SNPs (R^2^ ≥ 0.8) for each common AML/T2D susceptibility SNP, along with their chromosomal location, correlated alleles and associated genes, as collected from LDLink tool [24] (May 2021).

	Proxy SNPs	Chr	Position	Alleles	R^2^	Correlated Alleles	Associated Genes
rs11709077	rs17036160	3	12329783	(C/T)	0.9844	G = C,A = T	*PPARG*
	rs2012444	3	12375956	(C/T)	0.9751	G = C,A = T	
	rs13064760	3	12369401	(C/T)	0.9751	G = C,A = T	
	rs150732434	3	12360884	(G/-)	0.9751	G = G,A = -	
	rs13083375	3	12365308	(G/T)	0.972	G = G,A = T	
	rs35000407	3	12351521	(T/G)	0.9539	G = T,A = G	
	rs4684847	3	12386337	(C/T)	0.9391	G = C,A = T	
	rs11712037	3	12344730	(C/G)	0.9379	G = C,A = G	
	rs35788455	3	12388908	(TTG/-)	0.9362	G = TTG,A = -	
	*rs1801282*	3	12393125	(C/G)	0.9334	G = C,A = G	
	rs71304101	3	12396913	(G/A)	0.9083	G = G,A = A	
	rs35408322	3	12360357	(-/T)	0.9021	G = -,A = T	
rs1801282	rs4684847	3	12386337	(C/T)	0.9939	C = C,G = T	*PPARG, TIMP4*
	rs35788455	3	12388908	(TTG/-)	0.9908	C = TTG,G = -	
	rs71304101	3	12396913	(G/A)	0.9613	C = G,G = A	
	rs150732434	3	12360884	(G/-)	0.9573	C = G,G = -	
	rs13064760	3	12369401	(C/T)	0.9573	C = C,G = T	
	rs2012444	3	12375956	(C/T)	0.9573	C = C,G = T	
	rs13083375	3	12365308	(G/T)	0.9543	C = G,G = T	
	rs35000407	3	12351521	(T/G)	0.9365	C = T,G = G	
	*rs11709077*	3	12336507	(G/A)	0.9334	C = G,G = A	
	rs17036160	3	12329783	(C/T)	0.9183	C = C,G = T	
	rs35408322	3	12360357	(-/T)	0.8855	C = −,G = T	
	rs11712037	3	12344730	(C/G)	0.8806	C = C,G = G	
rs6685701	rs4970486	1	26871669	(C/T)	0.9826	A = C,G = T	*RPS6KA1*
	rs737465	1	26862939	(T/C)	0.9814	A = T,G = C	
	rs11580180	1	26867453	(A/G)	0.9746	A = A,G = G	
	rs2278978	1	26873245	(A/G)	0.9311	A = A,G = G	
	rs4443935	1	26875433	(A/G)	0.9072	A = A,G = G	
	rs10902750	1	26876245	(G/T)	0.9052	A = G,G = T	
	rs389548	1	26891697	(C/A)	0.8777	A = C,G = A	
rs11108094	rs11108087	12	95915763	(A/G)	0.8578	C = A,A = G	*USP44, METAP2*
	rs61939481	12	95921998	(T/C)	0.8477	C = T,A = C	
	rs143400372	12	95923620	(-/A)	0.8477	C = -,A = A	
	rs11108086	12	95914758	(T/C)	0.8187	C = T,A = C	
	rs79067108	12	95881761	(CT/-)	0.8141	C = CT,A = -	
	rs11108070	12	95881787	(T/A)	0.8141	C = T,A = A	
	rs12369757	12	95888603	(G/A)	0.8141	C = G,A = A	
	rs11108072	12	95890218	(T/C)	0.8141	C = T,A = C	
	rs10859889	12	95890413	(A/T)	0.8141	C = A,A = T	
	rs11522874	12	95893609	(G/A)	0.8141	C = G,A = A	
	rs61939476	12	95894581	(A/C)	0.8141	C = A,A = C	
	rs11108076	12	95897348	(G/A)	0.8141	C = G,A = A	
	rs11108079	12	95899173	(G/A)	0.8141	C = G,A = A	
	rs12146719	12	95901434	(C/A)	0.8141	C = C,A = A	
	rs61939479	12	95905364	(C/T)	0.8141	C = C,A = T	
	rs2305293	12	95879734	(C/T)	0.8095	C = C,A = T	
	rs11519597	12	95894247	(T/C)	0.8095	C = T,A = C	
	rs61939477	12	95896692	(A/G)	0.8095	C = A,A = G	
	rs4762563	12	95915341	(G/C)	0.805	C = G,A = C	
rs7929543	rs11603576	11	49344126	(G/A)	0.9947	A = G,C = A	*FOLH1, AC118942.1*
	rs10839264	11	49356806	(C/T)	0.9511	A = C,C = T	
	rs76218798	11	49356186	(T/C)	0.9366	A = T,C = C	
	rs11607791	11	49358347	(T/C)	0.9339	A = T,C = C	
	rs1880436	11	49344775	(A/G)	0.92	A = A,C = G	
	rs148517532	11	49332611	(A/G)	0.9188	A = A,C = G	
	rs144550850	11	49366641	(T/C)	0.9175	A = T,C = C	
	rs1843629	11	49319195	(G/A)	0.9161	A = G,C = A	
	rs75781920	11	49371482	(T/G)	0.9152	A = T,C = G	
	rs76427006	11	49375021	(T/A)	0.9149	A = T,C = A	
	rs7932396	11	49299282	(A/G)	0.9112	A = A,C = G	
	rs1843628	11	49323039	(A/G)	0.9033	A = A,C = G	
	rs7939300	11	49311134	(C/A)	0.8985	A = C,C = A	
	rs7939316	11	49311208	(A/G)	0.8985	A = A,C = G	
	rs11040313	11	49299786	(A/G)	0.8915	A = A,C = G	
	rs11040291	11	49248150	(C/T)	0.8898	A = C,C = T	
	rs61350355	11	49292311	(G/A)	0.8757	A = G,C = A	
	rs16906190	11	49203487	(A/G)	0.8709	A = A,C = G	
	rs11040354	11	49409798	(G/A)	0.847	A = G,C = A	
	rs10839244	11	49263085	(A/G)	0.8406	A = A,C = G	
	rs74380550	11	49236977	(C/T)	0.8301	A = C,C = T	
	rs59386222	11	49235409	(G/A)	0.8288	A = G,C = A	
	rs4091958	11	49234514	(T/C)	0.8286	A = T,C = C	
	rs11040365	11	49448078	(C/A)	0.826	A = C,C = A	
	rs10839237	11	49215635	(C/T)	0.8187	A = C,C = T	
	rs76002284	11	49271829	(A/G)	0.8145	A = A,C = G	
	rs11040352	11	49395272	(A/C)	0.8039	A = A,C = C	

**Table 4 ijms-22-09322-t004:** Common genes with common or different disease susceptibility SNPs for AML and T2D, as analyzed using data downloaded from the NHGRI-EBI Catalog of human GWAS [3] (May 2021).

	Gene Symbol	Full Gene Name	AML SNPs	T2D SNPs
1	*AC003681.1*	*-*	rs3788418, rs12627929, rs8139217, rs7285751, rs737903, rs36600, rs5752972, rs11090584, rs36608, rs5763609, rs39713, rs2051764, rs9614125, rs9625870, rs737904, rs737911, rs41170, rs5763681, rs36605, rs41158, rs4823058, rs41164, rs3788421, rs713718, rs5763559, rs737909, rs41159, rs3788425, rs5763688, rs7284538, rs5997546	rs41278853
2	*AC006041.1*	*-*	rs13225661, rs10242655, rs12113983, rs17348974, rs7811500, rs12532826, rs17169090, rs10950583	rs38221
3	*AC010967.1*	*-*	rs10204358, rs903230, rs745685, rs17044784, rs9677678, rs985549, rs903229, rs17044786, rs903231, rs17044787	rs9309245
4	*AC016903.2*	*-*	rs1545378	rs4482463
5	*AC022414.1*	*-*	rs10942819, rs10061629, rs6453303, rs11750661, rs17671389, rs9293712, rs9784696, rs6453304	rs7732130, rs4457053, rs6878122
6	*AC022784.1*	*-*	rs17656706, rs330003, rs6984551, rs11777846, rs75527, rs17149618, rs330035, rs330033, rs17656431, rs735449	rs17662402
7	*AC034195.1*	*-*	rs11717189, rs6768756	rs9842137
8	*AC069157.2*	*-*	rs10204358, rs903230, rs745685, rs17044784, rs9677678, rs985549, rs903229, rs17044786, rs903231, rs17044787	rs9309245
9	*AC073176.2*	*-*	rs950718	rs827237
10	*AC087311.2*	*-*	rs12227331, rs11052394	rs10844518, rs10844519
11	*AC093675.1*	*-*	rs4567941	rs34589210
12	*AC093898.1*	*-*	rs1503886, rs1039539, rs7673064, rs7681205, rs11934728, rs2320289, rs1847400, rs11941617	rs2169033
13	*AC097634.4*	*-*	rs9844845, rs17653411, rs9840264	rs844215, rs853866
14	*AC098588.2*	*-*	rs11100859, rs2719340, rs6817612	rs200995462
15	*AC098588.3*	*-*	rs11100859, rs2719340, rs6817612	rs200995462, rs75686861
16	*AC098650.1*	*-*	rs6549877, rs1350867, rs2371341, rs6549876, rs4258916, rs1381392, rs1563981, rs6549878	rs9869477
17	*AC114971.1*	*-*	rs10067455	rs73167517
18	*AC118942.1*	*-*	rs10501324, rs7929543, rs7115281, rs3960835, rs1164681, rs1164673, rs1164666, rs10769572, rs12806588, rs2204366, rs7930322, rs2205020, rs11040338, rs11040339, rs10839257, rs7118379, rs598101, rs10839272, rs7925896, rs7924782, rs7114817, rs588295	rs7929543
19	*AFF3*	*AF4/FMR2 Family Member 3*	rs6707538, rs7423759, rs17023314, rs4449188, rs7577040, rs17436893	rs34506349
20	*AL135878.1*	*-*	rs10138733, rs4981687, rs8016028, rs8022374, rs1951540, rs17114593, rs3950100, rs8022457, rs8016946, rs17560052, rs8020665	rs8005994
21	*AL135923.2*	*-*	rs10815796, rs10815795, rs10815793	rs10758950
22	*AL136114.1*	*-*	rs2065140, rs1885645, rs3131325, rs1923640, rs2065141, rs10494504, rs1885644	rs532504, rs539515
23	*AL136962.1*	*-*	rs7552571	rs9316706
24	*AL359922.1*	*-*	rs10965197, rs2027938, rs10757261, rs9657608	rs1063192
25	*AL391117.1*	*-*	rs10811816, rs10811815, rs1350996	rs11793831, rs7029718
26	*ASAH1*	*N-Acylsphingosine Amidohydrolase (Acid Ceramidase)*	rs17692377, rs382752, rs11782529	rs34642578
27	*AUTS2*	*Activator of Transcription and Developmental Regulator*	rs7459368, rs7791651, rs2057913, rs1557970, rs4718971, rs3922333, rs1008584, rs11772435, rs17578487, rs2057914, rs2057911, rs10486866	rs2103132, rs6947395, rs6975279, rs12698877, rs10618080, rs610930
28	*CACNA2D3*	*Calcium Voltage-Gated Channel Auxiliary Subunit Alpha2delta3*	rs11711040, rs6805548	rs76263492
29	*CHMP4B*	*Charged Multivesicular Body Protein 4B*	rs2050209, rs6088343, rs2092475, rs17091328	rs7274168
30	*CPNE4*	*Copine 4*	rs3851353, rs1010900, rs17341291, rs1850941, rs16838814, rs3900591, rs9853646, rs16838856, rs10512856, rs12636272, rs6792708, rs11708369, rs1505811, rs4522813, rs3914303, rs2369466, rs3922808, rs10934990, rs9876304, rs7626343	rs9857204, rs1225052
31	*CRTC1*	*CREB-regulated transcription coactivator 1*	rs2023878, rs17757406, rs6510997, rs12462498, rs6510999, rs2240887, rs7256986	rs10404726
32	*CSMD1*	*CUB and Sushi Multiple Domains 1*	rs592700, rs11779410, rs13277378, rs4876060, rs596332, rs673465	rs117173251
33	*DGKB*	*Diacylglycerol Kinase Beta*	rs10244653, rs10486042, rs17167995	rs17168486, rs10281892, rs11980500
34	*EIF2S2P7*	*Eukaryotic Translation Initiation Factor 2 Subunit Beta*	rs2193632, rs6714162, rs2870503, rs768329	rs1116357
35	*EML6*	*EMAP-Like 6*	rs10496035, rs4625954, rs13394146	rs5010712
36	*ERBB4*	*Erb-B2 Receptor Tyrosine Kinase 4*	rs10207288, rs10174084, rs13019783, rs4673628, rs4423543, rs6759039	rs3828242, rs13005841
37	*FAM86B3P*	*Family with sequence similarity 86, member A pseudogene*	rs13274039, rs2980417, rs2945230, rs2980422, rs10095669, rs2980420	rs7841082
38	*FSD2*	*Fibronectin type III and SPRY domain containing 2*	rs4779064	rs36111056
39	*GP2*	*Glycoprotein 2*	rs8046269, rs12930599, rs11642182, rs9937721, rs4383154	rs117267808
40	*GRID1*	*Glutamate Ionotropic Receptor Delta Type Subunit 1*	rs1991426, rs4933387, rs7084960, rs1896526, rs17096224, rs11201974, rs1896527, rs1896525, rs7918205	rs11201999, rs11201992
41	*GRK5*	*G Protein-Coupled Receptor Kinase 5*	rs12357403, rs17606601, rs4752269, rs10787945, rs7903013, rs12264832, rs17098576, rs12358835, rs12244897, rs10886439, rs4752276, rs17098586, rs10510056	rs10886471
42	*HPSE2*	*Heparanase 2*	rs12219674, rs527822, rs592142, rs10748739, rs657442, rs537851, rs521390, rs10883130, rs650527, rs526877, rs7907389, rs551674, rs10509724, rs523205, rs10883134, rs558398, rs526698, rs2018085, rs17538604, rs621644, rs552644, rs489611, rs552436, rs625777, rs11189692, rs563937, rs660426, rs17459507, rs898892, rs541519	rs524903
43	*KCNB2*	*Potassium Voltage-Gated Channel Subfamily B Member 2*	rs2251899	rs349359
44	*KCNQ1*	*Potassium Voltage-Gated Channel Subfamily Q Member 1*	rs10832134, rs12576156, rs11523905	rs2283159, rs163184, rs2237896, rs2283228, rs2237897, rs2237892, rs2237895, rs231362, rs2283220, rs231361, rs231349, rs163182, rs233450, rs77402029, rs2106463, rs463924, rs231356, rs233449, rs8181588, rs234853
45	*LCORL*	*Ligand-Dependent Nuclear Receptor Corepressor-Like*	rs1503886, rs1039539, rs7673064, rs7681205, rs11934728, rs2320289, rs1847400, rs11941617	rs2169033, rs2011603
46	*LDLRAD4*	*Low-Density Lipoprotein Receptor Class A Domain Containing 4*	rs7241766, rs6505821, rs7230189, rs8091352, rs7230276	rs11662800
47	*LHFPL3*	*LHFPL Tetraspan Subfamily Member 3*	rs2106504, rs17136882, rs13234807, rs6958831, rs7794181, rs979522, rs7787976, rs7787988	rs73184014
48	*LINC00424*	*Long Intergenic Non-Protein Coding RNA 424*	rs9316684, rs7320437, rs9316683, rs17074792	rs9316706
49	*LINC01234*	*Long Intergenic Non-Protein Coding RNA 1234*	rs4766686, rs10850140	rs7307263
50	*LINC02641*	*Long Intergenic Non-Protein Coding RNA 2641*	rs845083, rs2282015, rs1219960, rs845084, rs11597044, rs7091877, rs6599698	rs705145
51	*LINGO2*	*Leucine-Rich Repeat and Ig Domain Containing 2*	rs1452338, rs10511822, rs1349638, rs10124164, rs16912518	rs1412234
52	*MERTK*	*MER Proto-Oncogene, Tyrosine Kinase*	rs11684476	rs34589210
53	*MLIP*	*Muscular LMNA-Interacting Protein*	rs9357785, rs1325831, rs16884633, rs12191362, rs9464019, rs1359563, rs1325833, rs9637973, rs7750294, rs9370259	rs9370243
54	*MTMR3*	*Myotubularin-Related Protein 3*	rs3788418, rs12627929, rs8139217, rs7285751, rs737903, rs36600, rs5752972, rs11090584, rs36608, rs5763609, rs39713, rs2051764, rs9614125, rs9625870, rs737904, rs737911, rs41170, rs5763681, rs36605, rs41158, rs4823058, rs41164, rs3788421, rs713718, rs5763559, rs737909, rs41159, rs3788425, rs5763688, rs7284538, rs5997546	rs41278853
55	*NELL1*	*Neural EGFL-Like 1*	rs4412753, rs11025959, rs1377744, rs4923393, rs4576820, rs7119634, rs7948285, rs10500896, rs10833472, rs1945321	rs16907058
56	*NFATC2*	*Nuclear Factor of Activated T Cells 2*	rs17791950, rs4396773, rs4811167, rs6021170, rs1123479, rs959996	rs6021276
57	*NLGN1*	*Neuroligin 1*	rs9809489, rs6782940, rs16829698, rs1502461, rs6776485, rs16829573	rs686998, rs247975
58	*OARD1*	*O-Acyl-ADP-Ribose Deacylase 1*	rs6912013, rs9296355, rs7760860	rs7841082
59	*PAM*	*Peptidylglycine Alpha-Amidating Monooxygenase*	rs888801, rs467186, rs258132, rs462957, rs458256, rs2657459, rs401114, rs438126, rs451819, rs442443, rs382964, rs382946, rs647343	rs78408340
60	*PARD3B*	*Par-3 Family Cell Polarity Regulator Beta*	rs4673320, rs1990667, rs10179357, rs849207, rs16837235, rs907462, rs2160455, rs849250, rs12620034, rs10490293, rs10490292, rs4673324, rs4595957, rs4673329, rs2668152	rs4482463
61	*PCSK6*	*Proprotein convertase subtilisin/kexin type 6*	rs9806369, rs12905649, rs11858490, rs12719737, rs2047219, rs2047220, rs4965873, rs903552, rs11852310, rs11858491	rs6598475
62	*PKHD1*	*Polycystic kidney and hepatic disease 1*	rs1326570, rs41412044, rs9370050, rs728996, rs11754532, rs6458777, rs2104522, rs2894788, rs2397061, rs9474070, rs4715233, rs2104521, rs6922497, rs6940892-	rs1819564
63	*POLR1D*	*RNA Polymerase I And III Subunit D*	rs12584838, rs9551373, rs531950, rs10492484, rs7337722, rs667374, rs12876263, rs12870355, rs17821569, rs9507915, rs634035, rs542610, rs6491221, rs12050009	rs9319382
64	*PPARG*	*Peroxisome Proliferator Activated Receptor Gamma*	rs10517032, rs10517031, rs2324237, rs16874420, rs10020457, rs10517030, rs2324241	rs17036160
65	*PPP2R2C*	*Protein Phosphatase 2 Regulatory Subunit B gamma*	rs11946417, rs4505896, rs4689469, rs6446507, rs10937739, rs11938118, rs4689011, rs4689462, rs4076293, rs7654321, rs4234751, rs4689465	rs35678078
66	*PRAG1*	*PEAK1 Related, Kinase-Activating Pseudokinase 1*	rs13274039, rs2980417, rs2945230, rs2980422, rs10095669, rs2980420	rs7841082
67	*PTPRD*	*Protein Tyrosine Phosphatase Receptor Type D*	rs10815796, rs10815795, rs10815793	rs10758950, rs17584499
68	*RBMS3*	*RNA Binding Motif Single-Stranded Interacting Protein 3*	rs6549877, rs1350867, rs2371341, rs6549876, rs4258916, rs1381392, rs1563981, rs6549878	rs9869477
69	*RELN*	*Reelin*	rs6961175, rs10235204, rs2106283, rs2106282, rs6465955, rs6955789, rs6465954	rs39328
70	*RPL12P33*	*Ribosomal protein L12 pseudogene 33*	rs10774577, rs6489785, rs7300612, rs7969196, rs11065341, rs2701179, rs868795	rs118074491
71	*RPS6KA1*	*Ribosomal Protein S6 Kinase A1*	rs3127011, rs12094989, rs12723046, rs6685701, rs1982525, rs11576300, rs4659444, rs6670311	rs6685701
72	*RPTOR*	*Regulatory Associated Protein of MTOR Complex 1*	rs8065459, rs9915426, rs2333990, rs2589133, rs2138125, rs734338	rs11150745
73	*RREB1*	*Ras Responsive Element Binding Protein 1*	rs10458204, rs4960285, rs12196079, rs17142726, rs12197730, rs552188, rs7759330, rs3908470, rs6597246	rs9505085, rs9505097, rs9379084
74	*SEPTIN9*	*Septin 9*	rs8079522, rs1075457, rs3744069, rs9916143, rs312907, rs11658267, rs892961, rs566569, rs11650011, rs2411110	rs1656794
75	*SGCG*	*Sarcoglycan Gamma*	rs578196, rs501909, rs502068	rs9552911
76	*SGCZ*	*Sarcoglycan Zeta*	rs17608649, rs7826655, rs12547159, rs13278000	rs35753840, rs17294565
77	*SHROOM3*	*Shroom Family Member 3*	rs6848817, rs13151434, rs6810716, rs13105942, rs4241595, rs10050141, rs6854652	rs11723275, rs56281442
78	*SLC39A11*	*Solute Carrier Family 39 Member 11*	rs11077627, rs11077628, rs4530179, rs11658711	rs61736066
79	*SYT10*	*Synaptotagmin 10*	rs12227331, rs11052394	rs10844518, rs10844519
80	*TMEM106B*	*Transmembrane Protein 106B*	rs12537849, rs10237821, rs10269431, rs7794113	rs13237518
81	*TMEM87B*	*Transmembrane Protein 87B*	rs6713344, rs4848979, rs4848980	rs74677818
82	*TTN*	*Titin*	rs7604033, rs10497522, rs2291313, rs11902709, rs2291311, rs4894044, rs10497523, rs2054708, rs1484116, rs10171049, rs3754953, rs4471922, rs11895382, rs4894037, rs2291312, rs7600001	rs6715901
83	*USP44*	*Ubiquitin-Specific Peptidase 44*	rs3812813, rs10777699, rs2769444, rs7974458, rs10498964, rs301024, rs301003	rs2197973
84	*XYLT1*	*Xylosyltransferase 1*	rs4453460, rs4583225	rs551640889
85	*ZFHX3*	*Zinc Finger Homeobox 3*	rs328398, rs328389, rs328317, rs328384, rs328395	rs6416749, rs1075855
86	*ZNF800*	*Zinc Finger Protein 800*	rs11563463, rs2285337, rs2285338, rs11563346, rs11563634	rs17866443

**Table 5 ijms-22-09322-t005:** AML- or T2D- specific SNPs that act as eQTLs on the 86 common AML/T2D susceptibility genes in a tissue-specific manner, as analyzed via the GTex portal [21] (May 2021).

	AML-Specific			T2D-Specific		
	SNP ID	Associated Gene	Affected Gene (s)	SNP ID	Associated Gene	Affected Gene (s)
	Adipose, Muscle, Pancreas, Whole Blood					
1	rs1168446	*AC093675.1, MERTK*	*MERTK (ad, pa, bl), TMEM87B (mu, bl)*			
2	rs4848980	*TMEM87B*	*MERTK (pa, mu), TMEM87B (bl, ad)*			
3	rs5752972	*ASCC2, MTMR3*	*MTMR3 (ad, bl, mu, pa)*			
4	rs11684321	*MERTK*	*MERTK (pa, mu, ad, bl), TMEM87B (mu, ad, bl)*			
5	rs9625870	*ASCC2, MTMR3*	*MTMR3 (ad, bl, pa)*			
6	rs4848979	*TMEM87B*	*MERTK (pa, bl, mu, ad), TMEM87B (mu, pa, ad, bl)*			
7	rs1168446	*AC093675.1, MERTK*	*MERTK (pa, mu, ad), TMEM78B (ad, pa, mu, bl)*			
	Adipose, Muscle, Pancreas					
1	rs2769444	*USP44*	*USP44 (pa, mu, ad)*	rs4382480	*MFHAS1*	*FAM86B3P (ad, pa, mu), PRAG1 (ad),* *FAM85B (ad),*
2	rs13274039	*PRAG1, FAM86B3P*	*FAM86B3P (ad), FAM85B (ad)*			
3	rs301003	*USP44*	*USP44 (pa, mu, ad)*			
4	rs301026	*METAP2*	*USP44 (mu, pa, ad)*			
5	rs301024	*USP44*	*USP44 (pa, ad)*			
6	rs301009	*METAP2*	*USP44 (pa, mu, ad)*			
	Adipose, Muscle, Whole blood					
1	rs8139217	*MTMR3, AC003681.1*	*MTMR3 (bl, mu)*	rs7274168	*CHMP4B*	*CHMP4B (bl, mu, ad)*
2	rs737911	*MTMR3, AC003681.1*	*MTMR3 (ad, bl, mu)*			
3	rs7285751	*MTMR3, AC003681.1*	*MTMR3 (bl, mu, ad)*			
4	rs3788421	*MTMR3, AC003681.1*	*MTMR3 (bl, mu, ad)*			
5	rs41158	*HORMAD2-AS1, MTMR3, AC003681.1*	*MTMR3 (ad, bl, mu)*			
6	rs7284538	*MTMR3, AC003681.1*	*MTMR3 (bl, ad, mu)*			
7	rs41170	*HORMAD2-AS1, MTMR3, AC003681.1*	*MTMR3 (ad, bl, mu)*			
	Adipose, Pancreas, Whole blood					
1	rs4261758	*SPTBN1*	*EML6 (pa, ad, bl)*	rs34589210	*AC093675.1, MERTK*	*MERTK (pa), TMEM87B (ad, bl)*
2	rs4567941	*AC093675.1*	*MERTK (pa, bl), TMEM87B (ad, pa, bl)*			
3	rs36605	*MTMR3*	*MTMR3 (ad, bl, pa)*			
4	rs17039558	*TDRP*	*EML6 (pa, ad, bl)*			
5	rs737904	*MTMR3*	*MTMR3 (ad, bl, pa)*			
6	rs3811640	*MERTK*	*MERTK (pa), TMEM87B (ad, bl)*			
7	rs6734445	*SPTBN1*	*EML6 (pa, ad, bl)*			
8	rs36600	*MTMR3*	*MTMR3(ad, bl, pa)*			
9	rs11904679	*AC092839.1, SPTBN1*	*EML6 (pa, ad, bl)*			
10	rs6713344	*TMEM87B*	*MERTK (pa, bl, ad), TMEM87B (ad, pa, bl)*			
	Muscle, Pancreas, Whole blood					
1				rs13237518	*TMEM106B*	*TMEM106B (bl, pa, mu)*
	Adipose, Muscle					
1	rs11563634	*ZNF800*	ZNF800 (mu, ad)	rs11723275	*SHROOM3*	*SHROOM3 (mu, ad)*
2	rs10937739	PPP2R2C	PPP2R2C (mu, ad)			
3	rs2285338	*ZNF800*	*ZNF800 (ad, mu)*			
4	rs11563346	*ZNF800*	*ZNF800 (mu, ad)*			
5	rs4689465	*PPP2R2C*	*PPP2R2C (ad, mu)*			
6	rs4689469	*PPP2R2C*	*PPP2R2C (mu, ad)*			
	Adipose, Pancreas					
1	rs11887259	*MERTK*	*TMEM87B (ad), MERTK (pa, ad)*	rs7841082	*PRAG1, FAM86B3P*	*FAM86B3P (ad, pa), FAM85B (ad), PPP1R3B (pa)*
2	rs6729826	*SPTBN1*	*EML6 (ad)*			
3	rs4671956	*AC092839.2, SPTBN1*	*EML6 (ad, pa)*			
4	rs4374383	*MERTK*	*TMEM87B (ad), MERTK (pa, ad)*			
5	rs3811638	*MERTK*	*TMEM87B (ad), MERTK (pa, ad)*			
6	rs2945230	*PRAG1, FAM86B3P*	*FAM86B3P (ad, pa), FAM85B (ad)*			
7	rs13016942	*SPTBN1*	*EML6 (ad, pa)*			
8	rs12104998	*AC092839.1, SPTBN1*	*EML6 (ad, pa)*			
9	rs12105792	*SPTBN1*	*EML6 (ad, pa)*			
10	rs1367295	*AC092839.1, SPTBN1*	*EML6 (ad, pa)*			
11	rs11683409	*MERTK*	*MERTK (ad, pa), TMEM87B (ad)*			
12	rs17344072	*SPTBN1*	EML6 (ad, pa)			
	Adipose, Liver					
1	rs4659444	*DPPA2P2, HMGN2*	*RPS6KA1 (li)*			
2	rs1359563	*MLIP-AS1, MLIP*	*MLIP (ad, li)*			
3	rs12094989	*DPPA2P2, RPS6KA1*	*RPS6KA1 (li, ad)*			
4	rs9637973	*MLIP-AS1, MLIP*	*MLIP (li, ad)*			
5	rs1325831	*MLIP-AS1, MLIP*	*MLIP (li, ad)*			
	Adipose, Whole blood					
1	rs5997546	*ASCC2, MTMR3*	*MTMR3 (ad)*			
2	rs5763688	*MTMR3, AC003681.1*	*MTMR3 (ad, bl)*			
3	rs41159	*HORMAD2-AS1, MTMR3, AC003681.1*	*MTMR3 (ad, bl)*			
4	rs634035	*POLR1D*	*POLR1D (ad)*			
5	rs5763559	*ASCC2, MTMR3*	*MTMR3 (ad, bl)*			
6	rs737909	*MTMR3, AC003681.1*	*MTMR3 (ad, bl)*			
7	rs2051764	*MTMR3*	*MTMR3 (bl)*			
8	rs667374	*POLR1D*	*POLR1D (bl, ad)*			
	Muscle, Whole blood					
1	rs382752	*PCM1, ASAH1*	*ASAH1 (bl, mu)*			
	Pancreas, Whole blood					
1				rs74677818	*TMEM87B*	*TMEM87B (bl), MERTK (pa)*
	Adipose					
1	rs17821569	*POLR1D*	*POLR1D (ad)*	rs11201992	*GRID1*	*GRID1 (ad)*
2	rs12905649	*PCSK6*	*PCSK6 (ad)*	rs56281442	*SHROOM3*	*SHROOM3 (ad)*
3	rs10883130	*HPSE2*	*HPSE2 (ad)*	rs11201999	*GRID1*	*GRID1 (ad)*
4	rs12876263	*POLR1D*	*POLR1D (ad)*			
5	rs898892	*HPSE2*	*HPSE2 (ad)*			
6	rs7907389	*HPSE2*	*HPSE2 (ad)*			
7	rs7337722	*POLR1D*	*POLR1D (ad)*			
8	rs737903	*MTMR3*	*MTMR3 (ad)*			
9	rs10748739	*HPSE2*	*HPSE2 (ad)*			
10	rs2980420	*PRAG1, FAM86B3P*	*FAM86B3P (ad)*			
11	rs650527	*HPSE2*	*HPSE2 (ad)*			
12	rs7750294	*MLIP-AS1, MLIP*	*MLIP (ad)*			
13	rs10883134	*HPSE2*	*HPSE2 (ad)*			
14	rs2018085	*HPSE2*	*HPSE2 (ad)*			
15	rs41164	*HORMAD2-AS1, MTMR3, AC003681.1*	*MTMR3 (ad)*			
16	rs621644	*HPSE2*	*HPSE2 (ad)*			
17	rs542610	*POLR1D*	*POLR1D (ad)*			
18	rs489611	*HPSE2*	*HPSE2 (ad)*			
	Muscle					
1	rs4505896	*PPP2R2C*	*PPP2R2C (mu)*	rs11150745	*RPTOR*	*RPTOR (mu)*
	Pancreas					
1	rs9370050	*PKHD1*	*PKHD1 (pa)*			
	Liver					
1	rs12191362	*MLIP-AS1, MLIP*	*MLIP (li)*			
2	rs16884633	*MLIP-AS1, MLIP*	*MLIP (li)*			
	Whole blood					
1	rs382964	*PAM*	*PAM (bl), PPIP5K2 (bl)*	rs115505614	*GIN1*	*PAM (bl), PPIP5K2 (bl)*
2	rs10179948	*MERTK*	*TMEM87B (bl)*	rs35658696	*PAM*	*PAM (bl), PPIP5K2 (bl)*
3	rs382946	*AC099487.2, PAM*	*PAM (bl), PPIP5K2 (bl)*	rs75432112	*AC011362.1*	*PAM (bl), PPIP5K2 (bl)*
4	rs258132	*PAM*	*PAM (bl), PPIP5K2 (bl)*	rs9319382	*AL136439.1, POLR1D*	*POLR1D (bl)*
5	rs401114	*PAM*	*PAM (bl, ad), PPIP5K2 (bl)*	rs610930	*AUTS2*	*AUTS2 (bl)*
6	rs442443	*AC099487.2, PAM*	*PAM (bl), PPIP5K2 (bl)*	rs7729395	*PAM*	*PAM (bl), PPIP5K2 (bl)*
7	rs462957	*PAM*	*PAM (bl), PPIP5K2 (bl)*			
8	rs6088343	*CHMP4B, TPM3P2*	*CHMP4B (bl)*			
9	rs458256	*PAM*	*PAM (bl), PPIP5K2 (bl)*			
10	rs451819	*AC099487.2, PAM*	*PAM (bl)*			
11	rs17098576	*GRK5*	*GRK5 (bl)*			
12	rs17692377	*PCM1, ASAH1*	*ASAH1 (bl)*			
13	rs10211152	*MERTK*	*TMEM87B (bl), MERTK (bl)*			
14	rs12050009	*POLR1D*	*POLR1D (bl)*			
15	rs11782529	*PCM1, ASAH1*	*ASAH1 (bl)*			
16	rs9551373	*POLR1D*	*POLR1D (bl)*			
17	rs10095669	*PRAG1, FAM86B3P*	*FAM86B3P (bl)*			
18	rs467186	*PAM*	*PAM (bl)*			
19	rs6142044	*PIGPP3, TPM3P2*	*CHMP4B (bl)*			
20	rs2657459	*AC099487.2, PAM*	*PAM (bl), PPIP5K2 (bl)*			
21	rs438126	*AC099487.2, PAM*	*PAM (bl), PPIP5K2 (bl)*			
22	rs647343	*AC099487.2, PAM*	*PAM (bl), PPIP5K2 (bl)*			

ad: Adipose, bl: whole blood, li: liver, mu: muscle, pa: pancreas.

**Table 6 ijms-22-09322-t006:** Selected pathways significantly regulated by the set of 86 AML/T2D susceptibility genes plus seven eGenes affected by the five common AML/T2D susceptibility genes and their proxies, as analyzed upon processing in the STRING and KEGG databases [25,26]. Pathway IDs and description, number of susceptibility genes involved, number of background genes, their names as well as statistics (strength, FDR and log_10_FDR) for each pathway are reported.

Term ID	Term Description	Observed Gene Count	Background Gene Count	Strength	FDR	*log*_10_FDR	Matching Proteins in the Network
hsa00240	Pyrimidine metabolism	16	100	1.57	3.17 × 10^−18^	17.50	POLR2C, POLR2I, TWISTNB, POLR3B, POLR1A, POLR2D, POLR2J, POLR3E, POLR2G, POLR1D, POLR2L, POLR3C, POLR2K, POLR3H, POLR3A, POLR1C
hsa00230	Purine metabolism	16	173	1.33	6.30 × 10^−15^	14.20	POLR2C, POLR2I, TWISTNB, POLR3B, POLR1A, POLR2D, POLR2J, POLR3E, POLR2G, POLR1D, POLR2L, POLR3C, POLR2K, POLR3H, POLR3A, POLR1C
hsa04150	mTOR signaling pathway	14	148	1.34	3.30 × 10^−13^	12.48	MAPK1, TSC2, LAMTOR5, RHEB, RRAGB, LAMTOR1, RPTOR, EIF4EBP1, LAMTOR4, MTOR, LAMTOR2, RRAGD, RRAGC, RPS6KA1
hsa04152	AMPK signaling pathway	8	120	1.19	1.56 × 10^−6^	5.81	TSC2, RHEB, PPARGC1A, PPARG, RPTOR, PPP2R2C, EIF4EBP1, MTOR
hsa04211	Longevity regulating pathway	7	88	1.26	3.02 × 10^−6^	5.52	TSC2, RHEB, PPARGC1A, PPARG, RPTOR, EIF4EBP1, MTOR
hsa01100	Metabolic pathways	20	1250	0.57	4.74 × 10^−6^	5.32	POLR2C, POLR2I, TWISTNB, POLR3B, XYLT1, POLR1A, POLR2D, POLR2J, POLR2G, POLR1D, POLR2L, POLR3C, POLR2K, POLR3H, HPSE2, POLR3A, POLR1C, ASAH1, MTMR3, DGKB
hsa04910	Insulin signaling pathway	7	134	1.08	3.31 × 10^−5^	4.48	MAPK1, TSC2, RHEB, PPARGC1A, RPTOR, EIF4EBP1, MTOR
hsa05231	Choline metabolism in cancer	6	98	1.15	6.93 × 10^−5^	4.16	MAPK1, TSC2, RHEB, EIF4EBP1, MTOR, DGKB
hsa04151	PI_3_K-Akt signaling pathway	9	348	0.77	2.60 × 10^−3^	3.59	MAPK1, TSC2, RHEB, RPTOR, PPP2R2C, EIF4EBP1, ERBB4, MTOR, RELN
hsa05221	Acute myeloid leukemia	3	66	1.02	2.41 × 10^−2^	1.62	MAPK1, EIF4EBP1, MTOR

## Data Availability

Not applicable.

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
