# Peer review of "Common Genetic Aberrations Associated with Metabolic Interferences in Human Type-2 Diabetes and Acute Myeloid Leukemia: A Bioinformatics Approach"

_ijms, 2021, doi:10.3390/ijms22179322_

Round 1

Reviewer 1 Report

The authors described overlapping common susceptibility SNPs between type 2 diabetes (T2D) and acute myeloid leukemia (AML) AML. The manuscript is interesting. However several aspects of it remain not well elucidated. My comments are below:

  1. Out of 5 SNPs found to be overlapping between T2D and AML, two SNPs map on PPARG Beside the important role of PPARG gene in cancers, little is known about its role in AML. Because of the unknown role in AML, it is hard to give a correct estimation of its impact.
  2. Have the authors define the frequency of the SNPs in general population?
  3. The authors should use a validation cohort to estimate the exact involvement of these SNPs in both AML and T2D.
  4. Some of the sentences of the discussion section do not really address the role of the specific SNPs in both diseases. One example is given by the ones linked to GATA3.
  5. Although some of the SNPs hit some of the genes involved in AML, their involvement is not as driver genes of AML initiation.
  6. Some figures are not legible (e.g., Figure 4 panels A and B, Figure 5)
  7. The font in the main text is often different. 

Author Response

Response to Comments from Reviewer 1

Comment #1. Out of 5 SNPs found to be overlapping between T2D and AML, two SNPs map on PPARG Beside the important role of PPARG gene in cancers, little is known about its role in AML. Because of the unknown role in AML, it is hard to give a correct estimation of its impact.

Response: We thank the reviewer for this comment; we agree that we need to mention that due to the limited knowledge about PPARG in AML, it is not possible to pinpoint the role of its SNPs in the disease. To this end, we added the following period in the Discussion section of the revised manuscript (lines 465-468): “Moreover, in the case of certain genes and their SNPs, i.e. those of PPARG and GATA3, their specific implication in AML and/or T2D development is not well documented. Therefore, it is yet hard to provide a plausible explanation regarding their possible impact as risk factors for AML in the ground of T2D”. By using this sentence, we aimed at addressing also Comment #4, which regards GATA3.

Comment #2. Have the authors define the frequency of the SNPs in general population?

Response: We thank the reviewer for this comment; we agree it is essential to describe the frequency of these five common SNPs in the general population. For that reason, we downloaded data from https://gnomad.broadinstitute.org/, and generated a new figure (Supplementary Figure 1), where one can be informed about the frequency of the SNPs in the general population, in specific populations, in males and females, and also per age group. The Figure consists of bar diagrams and an embedded table with this information. In the Results section of the manuscript (lines 163-164), we added the sentence: “In addition, information regarding their frequency in the general population is reported in Suppl. Figure 1.”

Comment #3. The authors should use a validation cohort to estimate the exact involvement of these SNPs in both AML and T2D.

Response: This comment is of high value, and we really thank the reviewer for that. We agree that data revealed by bioinformatics analysis need to be validated in human cohorts. Unfortunately, within the time given for submission of this manuscript it was not possible to collect and assess the appropriate number of patients with T2D, AML, T2D that developed AML, or controls, to extract safe conclusions about the incidence of these SNPs in the aforementioned groups. The fact that we are processing only in-silico data is one of the limitations of our study mentioned in the Discussion section (lines 462-463): “It should be noted, however, that the study has certain limitations, including that it exclusively analyzed in-silico data…”. However, we are now in the process of setting up a corresponding study to follow our bioinformatics approach and we hope we have publishable data soon. To highlight the need for a validation study, we have added the following sentence in the paragraph of future perspectives in the Discussion section (lines 473-475): “For example, the common susceptibility genes revealed can be evaluated for their potential to serve as prognostic biomarkers of AML development in cohorts of T2D individuals.”

Comment #4. Some of the sentences of the discussion section do not really address the role of the specific SNPs in both diseases. One example is given by the ones linked to GATA3.

Response: We thank the reviewer for this comment; we agree that we need to mention that the limited knowledge regarding specific SNPs (like those in GATA3) does not allow us to address their role in AML or T2D or AML in the ground of T2D. We state this limitation in the sentence added in the Discussion section of the revised manuscript (lines 465-468), where we also aimed at addressing Comment #1 regarding SNPs in PPARG: “Moreover, in the case of certain genes and their SNPs, i.e. those of PPARG and GATA3, their specific implication in AML and/or T2D development is not well documented. Therefore, it is yet hard to provide a plausible explanation regarding their possible impact as risk factors for AML in the ground of T2D”.

Comment #5. Although some of the SNPs hit some of the genes involved in AML, their involvement is not as driver genes of AML initiation.

Response: We thank the reviewer for this comment; indeed, these genes are not driver genes for AML. We clarify this in the sentence added in lines 468-471: “Lastly, it needs to be clarified that although some of the reported SNPs are associated with certain genes involved in AML (such as RPS6KA1 and METAP2), the latter are not consider driver genes for AML initiation.”

Comment #6. Some figures are not legible (e.g., Figure 4 panels A and B, Figure 5).

Response: We thank the reviewer for this important comment; we made corresponding editing to the figures to be legible. In addition to Figures 4 and 5, we also processed the rest of the Figures of the manuscript (1, 2, 3) for best results.

Comment #7. The font in the main text is often different.

Response: We thank the reviewer for this comment; we edited the manuscript, so the font is the same throughout the text.

Reviewer 2 Report

In this article, Kyriakou et al. have made a very comprehensive bioinformatics study for identification of genetic alterations linking acute myeloid leukemia (AML) and type-2 diabetes. The paper is well written and organized, and data are presented exhaustively. The very big limitation of this study is that has been conducted only on published data and lacks functional studies; however, limitations have been clearly stated at the end of the manuscript.

Some very minor comments are summarized below.

  1. Introduction should be shortened (especially lines 47-70). On line 77, is the risk significantly different from general population? Please clarify.
  2. Decimals should be indicated consistently (use European or American style, because in tables decimals are in the European style, while in the text the American).
  3. Table 5 requires abbreviations in the caption (please clearly state that ad means adipose, pa pancreas, and so on).
  4. Figure 5. Graphs can be divided in two columns. In this way, they can be bigger and more legible.
  5. In the discussion, please shorten and avoid points that are too much speculative (e.g., lines 351-357; lines 398-419; lines, 446-454). When not necessary, do not jump from mouse models to humans, and not reference too frequently to figures and tables (the reader should not jump from discussion to results).
  6. Conclusions should be more concise and direct to the point. Lines 506-513 could be removed as seem to repeat previous statements. Or just move limitations at the beginning of conclusions and then conclude.

Author Response

Response to Comments from Reviewer 2

Comment #1. Introduction should be shortened (especially lines 47-70). On line 77, is the risk significantly different from general population? Please clarify.

Response: We thank the reviewer for this comment. We have now shortened the Introduction section (779 words in the previous section; now 668). Regarding the risk of AML in T2D, we agree with the Reviewer that we need to mention whether this is significantly higher compared to that in the general population. Indeed, based on the study of Harding, J. L., et al, Cancer risk among people with type 1 and type 2 diabetes: disentangling true associations, detection bias, and reverse causation. Diabetes Care 2015, 38, (2), 264-70, the standard AML incidence ratio in T2D is significantly higher compared to the general population. We added this information in lines 69-71 of the Introduction of the revised manuscript: “…, it has been described that the standard incidence ratio in a cohort of 641 T2D individuals is 1,36 (95% CI: 1,26-1,47), significantly higher compared to the general population [12].”

Comment #2. Decimals should be indicated consistently (use European or American style, because in tables decimals are in the European style, while in the text the American).

Response: We thank the reviewer for this comment. We have now made changes, so decimals are indicated consistently throughout the text.

Comment #3. Table 5 requires abbreviations in the caption (please clearly state that ad means adipose, pa pancreas, and so on).

Response: We thank the reviewer for this comment, and we apologize for this omission. We have now added the appropriate abbreviations in the legend of Table 5.

Comment #4. Figure 5. Graphs can be divided in two columns. In this way, they can be bigger and more legible.

Response: We thank the reviewer for this important comment. We made the suggested changes to Figure 5 to be legible. In addition, we made changes to all the Figures of the manuscript for best results.

Comment #5. In the discussion, please shorten and avoid points that are too much speculative (e.g., lines 351-357; lines 398-419; lines, 446-454). When not necessary, do not jump from mouse models to humans, and not reference too frequently to figures and tables (the reader should not jump from discussion to results).

Response: We thank the reviewer for this comment. We have now shortened the Discussion section (2235 words in the previous section; now 1750), following also suggested changes.

Comment #6. Conclusions should be more concise and direct to the point. Lines 506-513 could be removed as seem to repeat previous statements. Or just move limitations at the beginning of conclusions and then conclude.

Response: We thank the reviewer for this comment. We made changes also to the conclusion paragraph, following suggested changes.

Round 2

Reviewer 1 Report

The authors have appropriately answered the queries.